# Covalent Organic Framework/Polyacrylonitrile Electrospun Nanofiber for Dispersive Solid-Phase Extraction of Trace Quinolones in Food Samples

**DOI:** 10.3390/nano12142482

**Published:** 2022-07-20

**Authors:** Jinghui Zhou, An Chen, Hongying Guo, Yijun Li, Xiwen He, Langxing Chen, Yukui Zhang

**Affiliations:** 1College of Chemistry, Tianjin Key Laboratory of Biosensing and Molecular Recognition, Nankai University, Tianjin 300071, China; 2120200750@mail.nankai.edu.cn (J.Z.); 1120180234@mail.nankai.edu.cn (A.C.); 1120190301@mail.nankai.edu.cn (H.G.); yijunli@nankai.edu.cn (Y.L.); xiwenhe@nankai.edu.cn (X.H.); ykzhang@dicp.ac.cn (Y.Z.); 2National Demonstration Center for Experimental Chemistry Education, Nankai University, Tianjin 300071, China; 3Dalian Institute of Chemical Physics, Chinese Academy of Sciences, Dalian 116023, China

**Keywords:** electrospun nanofiber, covalent organic frameworks, dispersive solid-phase extraction, quinolones, food samples

## Abstract

The extraction of quinolone antibiotics (QAs) is crucial for the environment and human health. In this work, polyacrylonitrile (PAN)/covalent organic framework TpPa–1 nanofiber was prepared by an electrospinning technique and used as an adsorbent for dispersive solid-phase extraction (dSPE) of five QAs in the honey and pork. The morphology and structure of the adsorbent were characterized, and the extraction and desorption conditions for the targeted analytes were optimized. Under the optimal conditions, a sensitive method was developed by using PAN/TpPa–1 nanofiber as an adsorbent coupled with high-performance liquid chromatography (HPLC) for five QAs detection. It offered good linearity in the ranges of 0.5–200 ng·mL^−1^ for pefloxacin, enrofloxacin, and orbifloxacin, and of 1–200 ng·mL^−1^ for norfloxacin and sarafloxacin with correlation coefficients above 0.9946. The limits of detection (S/N = 3) of five QAs ranged from 0.03 to 0.133 ng·mL^−1^. The intra-day and inter-day relative standard deviations of the five QAs with the spiked concentration of 50 ng·mL^−1^ were 2.8–4.0 and 3.0–8.8, respectively. The recoveries of five QAs in the honey and pork samples were 81.6–119.7%, which proved that the proposed method has great potential for the efficient extraction and determination of QAs in complex samples.

## 1. Introduction

As natural or synthetic compounds, antibiotics have been widely used in human and veterinary medicine and can inhibit the proliferation of bacteria [1]. Quinolone antibiotics (QAs) are a typical class of synthetic antibiotics that are used in human medicine, animal husbandry, and animal feed to treat and prevent bacterial infections due to their broad antimicrobial spectrum, good oral absorption, and low cost [2]. However, intensive use of antibiotics in medicine has led to serious problems that their residues in honey, pork, milk, and other animal-derived foods are a potential source for a significant increase in allergic reactions or antibiotic resistance [3,4]. Hence, different countries and organizations such as the European Union and China have stipulated the maximum residue limits (MRLs) of QAs in foods [5,6]. Accordingly, it is of great importance to develop a sensitive and reliable method for the determination of quinolone antibiotic residues in animal-derived foods.

To date, a variety of detection methods, such as fluorescence [7], surface-enhanced Raman scattering (SERS) [8], chemiluminescence detection [9], capillary electrophoresis (CE) [10], and high-performance liquid chromatography (HPLC), equipped with a variety of detectors, including ultraviolet (UV) [11], diode array detection (DAD) [12], fluorescence (FLD) [13,14], and tandem mass spectrometry (MS/MS) [15,16], have been employed for the determination of QAs. Among these analytical techniques, HPLC-UV instrumentation has been widely used in routine analysis in analytical laboratories. However, there are still many challenges for measuring trace QAs in food and environmental matrices because these samples are complex and may contain substantial amounts of interfering substances that can influence detection and contribute to pollution on instruments. Therefore, sample pretreatment is often required before instrument analysis.

In recent years, some sample pretreatment techniques, such as conventional liquid–liquid extraction (LLE) [17,18], solid-phase adsorption-based extraction techniques such as solid-phase extraction (SPE) [19,20,21], solid-phase microextraction (SPME) [22,23], stir-bar sorptive extraction (SBSE) [24], dispersive solid-phase extraction (dSPE) [25,26], and magnetic solid-phase extraction (MSPE) [11,27,28,29], have been successfully applied to purification and separation of QAs from the complex samples. Numerous types of adsorbents, such as molecularly imprinted polymers, carbon materials, metal-organic frameworks, and covalent organic frameworks (COFs), have been utilized as the adsorbing materials for the extraction of QAs [30]. Thus, the development of new adsorbents with enhanced stability and efficiency on the extraction performance of the analytes is still a popular research topic. COFs have become a research hotspot.

COFs, as a new kind of porous organic material, are constructed by the connection of organic monomers via strong covalent bonds, which have attracted a wide range of attention of researchers [31]. Due to their tunable pore structure, large specific surface areas, and good thermal and chemical stability, COF materials exhibited great potential application in the extraction of analytes from complex samples [32,33,34]. However, the application of COFs in the extraction process would face tedious operations and bring unnecessary loss of adsorbents and analytes because small-sized COFs with powders often tend to easily leak during use. Therefore, combining covalent organic frameworks with other substrates to prepare composite materials could solve the shortcomings of COFs as adsorbents. Electrospinning technology makes polymer solutions or melts charged and deformed by high-voltage static electricity. When the charge repulsion force on the droplet surface exceeds the surface tension, the droplet will change. In the beginning, the droplet will enter a jet state, and the jet will stretch at a high speed in a short distance under the action of the electric field force. The solvent will volatilize and solidify, forming a nanofiber [35]. Therefore, combining covalent organic frameworks with electrospinning to prepare an electrospun nanofiber can avoid leakage and blockage of COF materials and is easy to operate. Yan et al. prepared electrospun PAN@SNW-1 and used it as a pipette tip solid-phase extraction (PT-SPE) adsorbent, which effectively extracted sulfonamides residues from meat samples [36]. Wang et al. used electrospun PAN@SCU1 to extract tetracycline antibiotics from ducks and grass carp [37]. Tian et al. fabricated electrospun covalent organic framework/polyacrylonitrile nanofiber as an adsorbent for the determination of plant growth regulators [38]. In our previous work, PAN/Tp-BD nanofiber was also used as an adsorbent to extract seven sulfonamides from the food matrix [39]. We fabricated superhydrophobic PAN/Tp-BD and PAN/TAPB-TPA nanofiber membranes at room temperature and applied them to separate water-in-oil mixtures [40].

In this work, PAN/TpPa–1 nanofiber membrane was prepared by doping COFs into PAN polymer solution and subsequently electrospinning. The PAN/TpPa–1 nanofiber was used as a dSPE adsorbent for the extraction of five QAs from animal-derived foods. The main parameters influencing the extraction efficiency of QAs, such as the amount of PAN/TpPa–1, desorption solvent, desorption time, extraction time, ionic strength of sample solution, pH of sample solution, and the analytical performance of the monitoring QAs, were evaluated. A sensitive method was developed by using PAN/TpPa–1-based dSPE coupled with HPLC and a UV detector and was successfully applied to the analysis of five QAs in honey and pork samples.

## 2. Experimental Procedures

### 2.1. Reagents and Chemicals

The 1,3,5-Triformylphloroglucinol (Tp) was obtained from Yanshen Technology Co., Ltd. (Changchun, China). *P*-phenylenediamine (Pa–1), mesitylene, and polyacrylonitrile (PAN, Mw = 150,000) were purchased from Shanghai Macklin Biochemical Co., Ltd. (Shanghai, China). Pefloxacin (PEF), norfloxacin (NOR), enrofloxacin (ENR), and orbifloxacin (ORB) were obtained from Qingdao Tenglong Weibo Technology Co., Ltd. (Qingdao, China). Sarafloxacin (SAR) was obtained from Shanghai Yuanye Bio-Technology Co., Ltd. (Shanghai, China). Methanol, ammonia (25–28%, *wt*%), formic acid, acetonitrile 1,4-dioxane, and *N*, *N*-dimethylformamide (DMF) were purchased from Tianjin Concord Technology Co., Ltd. (Tianjin, China). Honey and pork samples were obtained from a local supermarket in Tianjin. Ultrapure water used for all experiments was purchased from Watsons Water (Guangzhou, China).

### 2.2. Characterization and HPLC Conditions

All nanofibers obtained in this work were fabricated by an NS Lab NanoSpider (Elmarco, Liberec, Czech Republic). Scanning electron microscopy (SEM) images were obtained with a SU3500 (Hitachi, Japan). All samples needed sputtering before observation, and the operating voltage was 30 kV. The FT-IR spectra of materials were obtained by FT-IR spectrometer (Bruker Tensor 27, Borken, Germany) in the range of 3500–500 cm^−1^. The X-ray diffraction (XRD) pattern was recorded in the range of 2θ from 2 to 35° with an X-ray diffractometer (Rigaku, Japan). The tube voltage was 40 kV, the tube current was 150 mA, and the scanning speed was 15 deg/min. The surface area and pore size were calculated by nitrogen adsorption and desorption isotherms at 77 K, according to the Brunauer–Emmett–Teller (BET) method (ASAP 2020, Micrometrics, Norcross, GA, USA). The thermogravimetric analysis (TGA) was performed on a TG209C (Netzsch, Selb, Germany). The temperature range was from 25 °C to 800 °C, and the heating rate was 10 °C/min.

The HPLC measurement was carried out by an LC-20AT (Shimadzu, Kyoto, Japan) HPLC system equipped with an SPD-20A UV/vis detector. An Inert Sustain C18 Column (250 × 4.6 mm, 5.0 μm) was used for separating five QAs. The mobile phase was water (containing 0.3% formic acid)–methanol (71:29, *v*/*v*), and the volumetric flow rate was set at 0.5 mL·min^−1^. The injection volume is 20 μL, and the detection wavelength was 277 nm for analytes. Five QAs were separated with the elution order of PEF, NOR, ENR, ORB, and SAR.

### 2.3. Preparation of TpPa–1

TpPa–1 was prepared according to the reported literature with slight modifications [41]. Typically, Tp (252 mg, 1.2 mmol), Pa–1 (192 mg, 1.8 mmol), and 12 mL of mesitylene/dioxane (1:1, *v*/*v*) were added into a glass bottle. The mixture was sonicated for 30 s to obtain a homogenous dispersion, and 2.0 mL of 3 M aqueous acetic acid was added. The mixture was transferred into a 50-mL Teflon-lined reactor. The reactor was sealed and then heated at 120 °C for 72 h. The product was washed with acetone and anhydrous ethanol and then dried at 80 °C under vacuum overnight, and a deep red-colored powder was collected.

### 2.4. Preparation of PAN/TpPa–1 Nanofiber

As shown in Figure 1, PAN/TpPa–1 nanofiber was prepared by needleless electrospinning. Typically, 176.0 mg of TpPa–1 was dispersed in 10 mL of DMF by sonicating to obtain a homogenous dispersion. Then, 1.0 g of PAN was added, and the mixture was stirred overnight to obtain a homogeneously dispersed PAN/TpPa–1 electrospinning precursor solution. The distance between the two electrode wires was 16 cm, and the voltage was 70 kV. The velocity of the vessel loaded with polymer solution was 50 mm/s. The relative humidity of the whole electrospinning process was kept at 30–40%. The prepared PAN/TpPa–1 nanofiber was dried at 60 °C overnight in a vacuum oven. The preparation of pure PAN nanofiber followed the same procedure except for the addition of TpPa–1 powders.

### 2.5. Samples Preparation

A 10.0 g honey sample was vortexed with 30 mL of acetonitrile for 2 min. Then, the mixture was sonicated for 30 min. After that, the sample was centrifuged at 6000 r/min for 7 min, and the sediment was repeatedly extracted with 30 mL of acetonitrile once. The above extraction steps were repeated for five groups. The supernatants obtained were combined to evaporate dry with a rotary evaporator, and the residues were dissolved with 5 mL of acetonitrile. After being filtered by a 0.22-µm filter membrane, the obtained solution was diluted to 50 mL with ultrapure water and stored in a refrigerator at 4 °C until use. 

The pork was cut into small pieces. A 10.0 g pork sample was vortexed with 30 mL of 1% acetic acid-acetonitrile for 5 min. Then, the sample was centrifuged at 3500 r/min for 5 min. The supernatant was collected, and the residues were extracted again. The above extraction steps were repeated for five groups. The supernatant obtained was combined. Then, 30 mL of n-hexane was added into the solutions mentioned above. The solution was vortexed for 1 min and immediately centrifuged at 3500 r/min for 5 min. The n-hexane layer was discarded, and the acetonitrile layer was collected. Afterwards, 10 mL of isopropanol was placed in the collected acetonitrile layer with adequate mixing. The above solution was evaporated until dry on a rotary evaporator, and the residues were dissolved in 5 mL of acetonitrile. The next step is similar to the honey sample. 

### 2.6. dSPE Procedure

The dSPE procedure utilizing the PAN/TpPa–1 as an adsorbent was as follows. First, 10.0 mg of PAN/TpPa–1 was placed in a 50-mL centrifugal tube and activated with 2.0 mL of methanol and 2.0 mL of water for 1 min in turn. Then, 20 mL of the sample solution was loaded into the centrifugal tube, and the mixture was shaken for 40 min. Thereafter, the PAN/TpPa–1 was washed with 2.0 mL of water, and the adsorbed target analytes were eluted with 1.0 mL of 2.0% ammonia-methanol (*v*/*v*) by shaking for 15 min. After being filtered by a 0.22-µm membrane, the eluent was concentrated until dry using nitrogen. Finally, the residues were redissolved with 50 µL of eluent for HPLC analysis. The dSPE procedure is shown in Figure 1.

## 3. Results and Discussions

### 3.1. Characterization of PAN/TpPa–1 Nanofiber

The morphologies of TpPa–1, pure PAN, and PAN/TpPa–1 nanofiber were investigated by SEM, and the images are shown in Figure 2. It can be seen that the TpPa–1 COF particles gathered and have a micrometer size (Figure 2A). There may be some problems, such as adsorbent leakage, equipment blockage, and difficulty to recycle, when pure COFs were used as an adsorbent. Therefore, PAN nanofiber was chosen as supporting substrates for COF in this work. As shown in Figure 2B, the surface of pure PAN nanofiber was smooth, and the sizes of PAN nanofiber were mainly distributed between 180 nm and 220 nm (Appendix A), which indicated that the fiber sizes were relatively uniform. The SEM images of PAN/TpPa–1 nanofiber by simply blending COF into PAN nanofiber showed that TpPa–1 was successfully doped into PAN (Figure 2C,D).

The FT-IR of Tp, Pa–1, and COF TpPa–1 is displayed in Figure 3. It can be seen that the N–H characteristic peak of Pa–1 completely disappeared in TpPa–1, indicating that Pa–1 was completely formed of COF TpPa–1. The characteristic peak of C=O in TpPa–1 exhibited a blue shift relative to Tp (1639 cm^−1^) and merged with the peak of the C=C stretching band at 1582 cm^−1^, which was due to the strong hydrogen bond in the keto structure. The peaks at 1256 cm^−1^ come from C–N stretching. This indicated that TpPa–1 was successfully synthesized and the tautomerism of TpPa–1 tended to form a keto structure instead of an enol structure. 

X-ray diffraction (XRD) analysis was used to study the crystal structures of TpPa–1, PAN, and PAN/TpPa–1 nanofiber. It can be seen in Figure 4 that the first intense peak of TpPa–1 appears at ~4.7° (*2θ*), corresponding to the (100) reflecting plane, and the minor peaks appear at ~8.1° and ~27° (*2θ*), attributable to the (200) and (001) planes, respectively. At the higher ~27° (*2θ*), it was mainly due to π–π stacking between COFs layers. This indicated that TpPa–1 has good crystallinity, which was consistent with the simulation results. The unit cell was constructed with a hexagonal model *P6/M* space group, and the unit cell parameters were *a* = *b* = 22.82 Å and *c* = 3.34 Å. After TpPa–1 was doped into PAN, there were no obvious peaks at ~4.7°, ~8.1°, and ~27° (*2θ*), and the XRD pattern was basically consistent with PAN, which may be due to the low content of TpPa–1 doped into PAN.

The surface area and pore size were calculated by nitrogen adsorption–desorption isotherms, according to the Brunauer–Emmett–Teller (BET) method. As shown in Appendix A, the N_2_ adsorption–desorption isotherm of TpPa–1 increased sharply when P/P_0_ was very low because the interaction between adsorbent and adsorbate was enhanced in narrow micropores, which led to micropore filling at extremely low relative pressures. However, the saturated pressure was reached, and the adsorbate condensed led to the curve rising. This was a typical type I adsorption isotherm of microporous materials, indicating that TpPa–1 was mainly composed of micropores with a pore diameter of about 1.7 nm and a specific surface area of 778.4 m^2^·g^−1^. Therefore, TpPa–1 is very suitable for the adsorption of five QAs whose sizes were between 1.24 nm and 1.55 nm (Appendix A). As shown in Appendix A, the isotherm of PAN nanofiber showed steep absorption at high relative pressure and an almost uniform adsorption–desorption isotherm at relative pressure lower than 0.8, while PAN/TpPa–1 nanofiber showed adsorption and desorption behavior between bare PAN nanofiber and COFs due to their binary composition. The BET specific surface areas of TpPa–1, PAN, and PAN/TpPa–1 were 778.4 m^2^·g^−1^, 12.5 m^2^·g^−1^, and 9.3 m^2^·g^−1^, respectively. This shows that most TpPa–1 clusters cannot adsorb nitrogen, and most TpPa–1 particles in PAN/TpPa–1 nanofiber were blocked by PAN.

The thermogravimetric analysis (TGA) of the materials is shown in Appendix A. TGA curves showed that TpPa–1 had a mass loss of 8% when the temperature reached 107 °C, which may be due to the evaporation of residual water and solvent. At 300 °C, there was an obvious mass loss. The material lost 57% of its mass at 780 °C. A rapid mass loss of PAN nanofiber began at 300 °C, but it slowed after 460 °C, and the mass loss reached 48% at 780 °C. The mass loss behavior of PAN/TpPa–1 combined the behaviors of PAN and TpPa–1 but was closer to that of PAN, indicating that there were binary components in PAN/TpPa–1, and the content of PAN was higher than TpPa–1.

### 3.2. Optimization of dSPE

#### 3.2.1. Effect of Amount of PAN/TpPa–1 Nanofiber on the Extraction of QAs

The amount of adsorbent is of great significance to the dSPE procedure. In this work, 4 mg, 7 mg, 10 mg, and 13 mg were selected to investigate the influence of the amount of adsorbent on the extraction of QAs. With the increase in adsorbent mass from 4 mg to 10 mg, the peak areas of the chromatogram for five QAs were greatly improved. When the adsorbent mass increased from 10 mg to 13 mg, the peak areas of the targets did not significantly increase; so, the amount of PAN/TpPa–1 nanofiber selected for further research was 10 mg (Figure 5A).

#### 3.2.2. Effect of Desorption Solvent and Desorption Time on the Extraction of QAs

Methanol, acetonitrile, ethyl acetate, and acetone were selected as eluents to investigate the effect of desorption solvent. The results showed that the target analytes were hardly eluted under the condition of these eluents. Considering that QAs are a kind of acid–base amphoteric compound, it may be necessary to add an appropriate acid or alkali to the eluant to improve the elution efficiency. Therefore, the desorption effects of 2% formic acid-methanol, 1% ammonia-methanol, 2% ammonia-methanol, and 4% ammonia-methanol were investigated, and the results are shown in Figure 5B. Adding an appropriate acid into the eluent was beneficial to the elution of five QAs, but the elution effect was inferior to adding an alkali. The optimal elution effect was obtained by adding 2% ammonia–methanol. The influence of elution time on the extraction is shown in Figure 5C. The elution effect for five QAs was improved along with the elution time from 2 min to 15 min. When the elution time was further increased from 15 min to 30 min, the elution effect of sarafloxacin and orbifloxacin was slightly improved, but the elution effect of norfloxacin decreased obviously to some extent. Therefore, 15 min was selected as the elution time for further experimentation.

#### 3.2.3. Effect of Extraction Time on the Extraction of QAs

During the process of dSPE, the extraction time is important. In this work, the effect of extraction time on the dSPE process was investigated. As shown in Figure 5D, the extraction efficiency was increased in the first 40 min and did not significantly improve when the extraction time increased from 40 min to 60 min. Therefore, 40 min was selected as the extraction time.

#### 3.2.4. Effect of Ionic Strength of Sample Solution

The influence of ionic strength on dSPE efficiency was also investigated by adding different amounts of sodium chloride into the sample solution. It was found that the extraction effect of five QAs was obviously reduced with the increase in sodium chloride concentration from 0 to 100 mmol·L^−1^ (Figure 5E). The reason for this may be that the addition of salt increased the viscosity of the solution, which led to a decrease in the mass transfer rate. It was not conducive to the extraction of the targets, thus leading to an obvious decrease in extraction efficiency. Therefore, NaCl was not added in subsequent experiments.

#### 3.2.5. Effect of pH Value of Sample Solution

The pH of the solution can affect the degree of protonation of the analytes, thus affecting the effect of extraction. The pH value of the sample solution was adjusted from 2 to 10 by adding HCl or NaOH. As shown in Figure 5F, the extraction efficiency remained consistent in the range of pH 4 to 8, but became worse at pH 2 and pH 10. Considering that the peak area of orbifloxacin among the five targets was small, and its extraction effect at pH 4 was better than at pH 8, pH 4 was selected as the optimized pH value.

### 3.3. Reproducibility and Reusability of PAN/TpPa–1 Nanofiber

The reusability of the adsorbent is an important index to evaluate the extraction efficiency in the dSPE process. In this work, the PAN/TpPa–1 nanofiber was washed with 3 mL eluent for 5 min, and the operation was repeated three times to ensure that there was no residue of the targets. The regenerated adsorbent was used in the next dSPE process. It was observed that the PAN/TpPa–1 nanofiber can still achieve a good extraction effect after regenerating six times (Appendix A). This showed that PAN/TpPa–1 nanofiber had good reusability.

### 3.4. Method Validation and Application in Real Sample

Under the optimal conditions, the validity of this method was proved by linear range, limits of detection (LODs), and limits of quantitation (LOQs). As shown in Table 1, for five targets, the linear ranges of pefloxacin, enrofloxacin, and orbifloxacin were 0.5–200 ng·mL^−1^, and the linear ranges of norfloxacin and sarafloxacin were 1–200 ng·mL^−1^ (R^2^ > 0.9946). The LODs (S/N = 3) and LOQs (S/N = 10) were in the range of 0.03–0.133 ng·mL^−1^ and 0.086–0.288 ng·mL^−1^, respectively. The intra-day RSD and inter-day RSD were 2.8–4.3% and 3.0–8.8%, respectively, which showed the method has good precision.

The practicability of this method was evaluated by using animal-derived food (honey and pork) as real samples and three spiked concentrations (5, 50, and 100 ng·mL^−1^) were investigated (Table 2). Five QAs were not detected in blank samples of honey and pork, and the recoveries in spiked honey samples and pork samples were 82.6–119.7% and 81.6–119.0% with relative standard deviations of 1.0–9.1% and 4.5–9.8%, respectively. The typical chromatograms of honey and pork samples are shown in Figure 6. The results showed that the method was reliable and suitable for the determination of QAs in animal-derived foods.

### 3.5. Comparison with Other Reported Methods 

To evaluate the performance of this method, it was compared with other methods for adsorbing QAs (Table 3). Through the comparison of LOD, recovery, elution volume, and amount of adsorbent, it can be seen that this method has the advantages of low LOD, smaller elution volume, less amount of adsorbent, and equivalent recovery to other methods. Therefore, the method based on PAN/TpPa–1–dSPE–HPLC can be used as an enrichment method for the determination of QAs in animal-derived foods.

## 4. Conclusions

In this work, a novel PAN/TpPa–1 nanofiber was synthesized by a facile electrospinning method and applied as a dSPE adsorbent for extraction of five trace QAs in honey and pork, combined with HPLC–UV detection. The prepared PAN/TpPa–1 material exhibited a good extraction effect towards QAs probably due to π–π interactions and hydrogen-bond formation. The established dSPE/HPLC–UV method exhibited a wide linear range, low LODs, and satisfactory recoveries. Furthermore, the PAN/TpPa–1 nanofiber possessed good reusability and easy operation. In addition, the results for the extraction of five QAs provide important reference information and an experimental basis. Therefore, the dSPE/HPLC–UV method based on PAN/TpPa–1 had excellent extraction performance for QAs and has application potential for the determination of QAs in complex samples.

## Figures and Tables

**Figure 1 nanomaterials-12-02482-f001:**
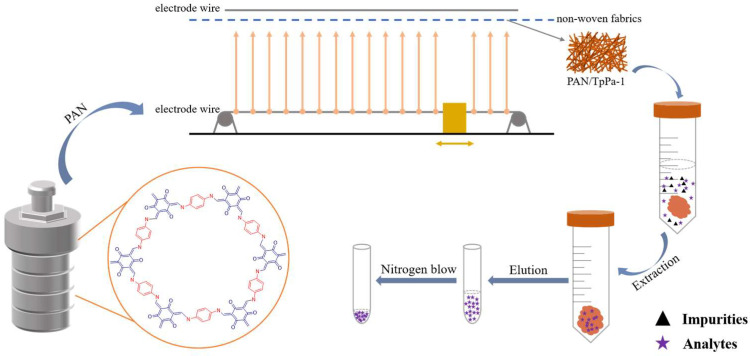
Schematic diagram of the preparation of PAN/TpPa–1 nanofiber and used as a dSPE adsorbent of QAs.

**Figure 2 nanomaterials-12-02482-f002:**
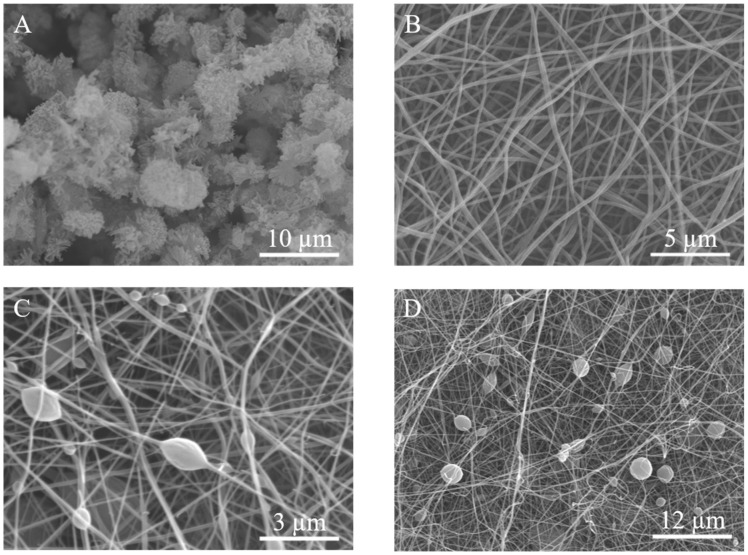
SEM images of TpPa–1 (**A**), pure PAN (**B**), PAN/TpPa–1 (**C**,**D**) nanofiber.

**Figure 3 nanomaterials-12-02482-f003:**
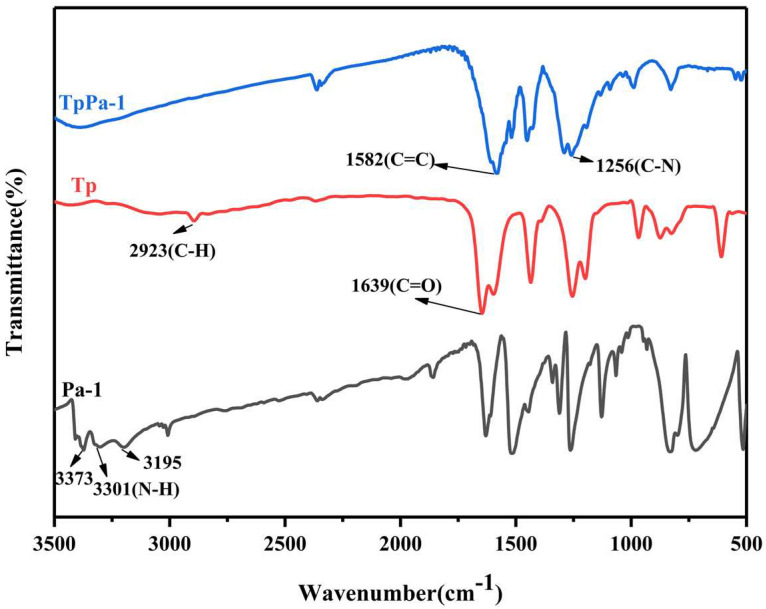
FT-IR spectra of Tp, Pa–1, and COF TpPa–1.

**Figure 4 nanomaterials-12-02482-f004:**
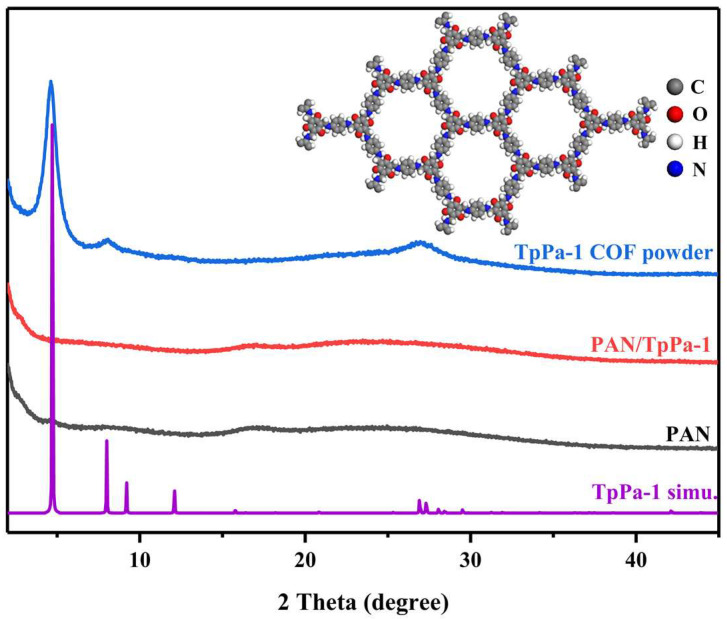
XRD patterns of TpPa–1, PAN/TpPa–1 nanofiber, pure PAN nanofiber, and simulation of TpPa–1.

**Figure 5 nanomaterials-12-02482-f005:**
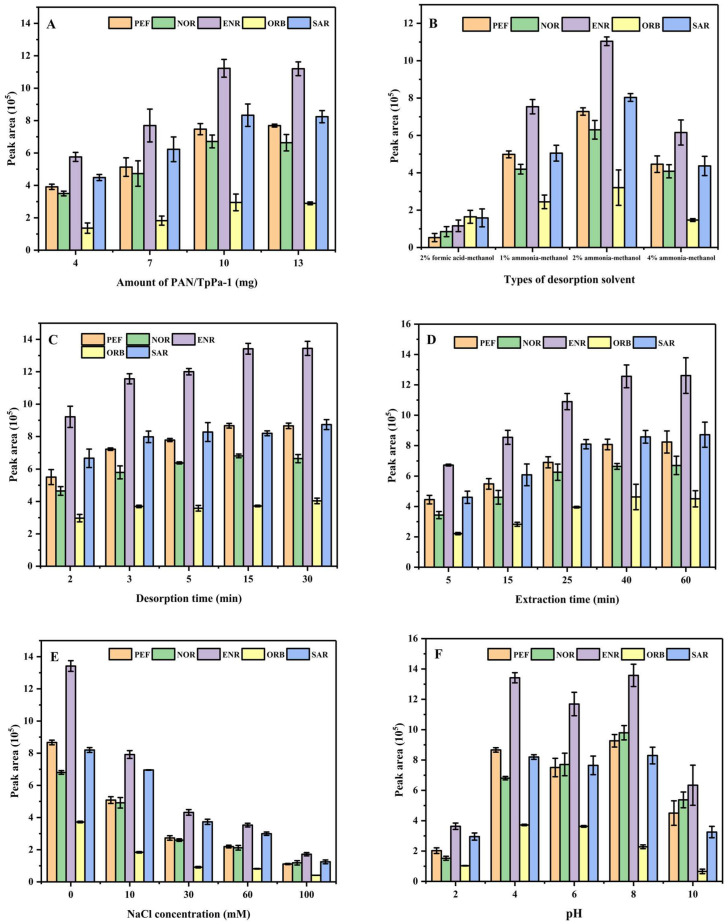
Effect of amount of PAN/TpPa–1 nanofiber (**A**), types of desorption solvent (**B**), desorption time (**C**), extraction time (**D**), concentration of NaCl (**E**), and standard solution pH value (**F**) on the extraction performance of QAs.

**Figure 6 nanomaterials-12-02482-f006:**
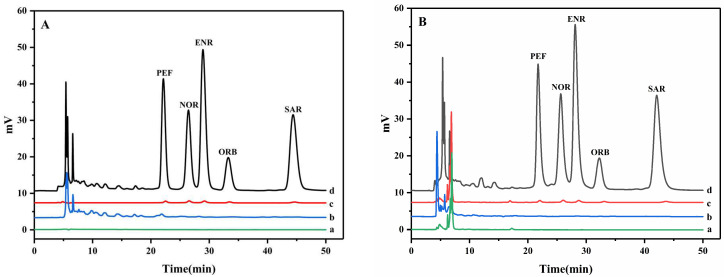
(**A**): HPLC-UV chromatograms of (a) honey sample; (b) honey sample after dSPE; (c) honey sample spiked with 100 ng·mL^–1^ before dSPE; and (d) honey sample spiked 100 ng·mL^–1^ after dSPE. (**B**): HPLC-UV chromatograms of (a) pork sample; (b) pork sample after dSPE; (c) pork sample spiked with 100 ng·mL^–1^ before dSPE; and (d) pork sample spiked 100 ng·mL^–1^ after dSPE.

**Table 1 nanomaterials-12-02482-t001:** Characteristic data for the dSPE with PAN/TpPa–1 nanofiber as sorbent for the subsequent HPLC determination.

Analytes	Linear Range (ng·mL^−1^)	R^2^	LODs (ng·mL^−1^)	LOQs (ng·mL^−1^)	RSDs (%) (*n* = 3) (50 ng·mL^−1^)	
					Intra-Day	Inter-Day
PEF	0.5–200	0.9946	0.039	0.110	4.0	4.8
NOR	1.0–200	0.9967	0.069	0.233	4.3	8.2
ENR	0.5–200	0.9972	0.030	0.086	2.8	3.0
ORB	0.5–200	0.9951	0.133	0.288	2.9	8.8
SAR	1.0–200	0.9974	0.073	0.226	3.9	6.3

**Table 2 nanomaterials-12-02482-t002:** The determination of five QAs in honey and pork samples (*n* = 3).

Analytes	Added (ng·mL^−1^)		Honey			Pork	
		Found (ng·mL^−1^)	Recovery (%)	RSDs (%)	Found (ng·mL^−1^)	Recovery (%)	RSDs (%)
PEF	0	ND	–	–	ND	–	–
	5	4.7	93.8	8.0	5.0	100.0	7.9
	10	55.7	111.3	6.4	42.9	85.7	5.6
	100	87.2	87.2	1.0	95.7	95.7	8.4
NOR	0	ND	–	–	ND	–	–
	5	5.6	112.6	5.8	5.9	119.0	9.2
	10	54.4	108.7	9.1	45.0	89.9	6.4
	100	82.7	82.7	3.0	101.9	101.9	6.2
ENR	0	ND	–	–	ND	–	–
	5	4.8	96.1	6.5	5.4	108.6	4.5
	10	55.3	110.6	2.6	44.1	88.3	5.6
	100	87.0	87.0	3.1	96.5	96.5	5.8
ORB	0	ND	–	–	ND	–	–
	5	4.1	82.6	2.7	4.4	88.9	6.9
	10	52.0	104.0	9.8	40.8	81.6	4.8
	100	83.7	83.7	6.7	86.3	86.3	4.8
SAR	0	ND	–	–	ND	–	–
	5	6.0	119.7	2.6	5.4	107.6	9.8
	10	53.7	107.4	8.3	44.6	89.1	8.6
	100	86.2	86.2	5.0	107.1	107.1	7.0

**Table 3 nanomaterials-12-02482-t003:** Comparison of some methods for the determination of QAs.

Method	Adsorbent	LOD (μg·kg^−1^ or μg·L^−1^)	Recovery (%)	Elution Volume	Amount of Adsorbent (mg)	Ref.
CE-UV	MMMIPs	12.9–8.8	92.7–108.6	–	30 mg	[42]
HPLC-UV	MIP@UiO-66-NH_2_	0.19–0.39	92.6–100.5	3 mL	20 mg	[43]
HPLC-DAD	MWCNTs-Fe_3_O_4_@SiO_2_-CS	1.5–3	81.2–109	5 mL	30 mg	[44]
HPLC-FD	MIM/C_3_N_4_	0.2–0.8	92.1–99.4	1 mL	30 mg	[45]
HPLC-UV	Fe_3_O_4_@MI-POSS	1.76–12.42	75.6–108.9	2 mL	60 mg	[46]
HPLC-MS/MS	Fe_3_O_4_@COF(TpBD)@Au-MPS	0.1–1.0	82–110.2	1 mL	10 mg	[27]
HPLC-DAD	Mag@GO-g-CNCs@MIPs	6.5–51	79.2–96.1	2 mL × 3	20 mg	[47]
HPLC-UV	PAN/TpPa–1	0.03–0.133	81.6–119.7	1 mL	10 mg	This work

## Data Availability

Not applicable.

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
