# Peer review of "Covalent Organic Framework/Polyacrylonitrile Electrospun Nanofiber for Dispersive Solid-Phase Extraction of Trace Quinolones in Food Samples"

_nanomaterials, 2022, doi:10.3390/nano12142482_

Round 1
Reviewer 1 Report
(i) Title of the manuscript could be further amended to reflect the novelty of the research conducted. As per now, it looks good but could be further improve to reflect the novelty.
(ii) An abstract is a shortened version of the work conducted and should contain all information necessary for the reader to determine. It would be great if the student could improve by adding those elements:
(1) what the objectives of the study were;
(2) how the study was done;
(3) what results were obtained;
(4) and the significance of the results.
(5) the application where the research could be implemented.
(iii) In term of clarity of presentation, it is to the acceptable level.
(iv) In term of organization which include list of introductions, methodology, results and analysis, references and etc, it all looks good. No further amendments needed.
(v) Manuscript needs to be sent for English Proofread.
(vi) There are some typos and spelling mistakes detected while reviewing the manuscript. This could be overcome by sending this manuscript for English Proofread.
(vii) Terminology operational definition used is at acceptable level.
(viii) Why is this study of scientific interest and the objectives should be rewording to have a link with one another?
(ix) This section discusses the results and conclusions of previously published studies, to help explain why the current study is of scientific interest.
(x) The Introduction is organized to move from general information to specific information. The background must be summarized concisely, but it should not be itemized. Limit the introduction to studies that relate directly to the present study. Emphasize your specific contribution to the topic.
(xi) The last sentences of the introduction should be an overall statement of objectives and a statement of hypotheses. This will be a good transition to the next section, Methods, in which you will explain how you proceeded to meet your objectives and test your hypotheses.
(xii) I am suggesting that the student should have a summary table that shows the GAP of the studies. This could be benefiting the readers to see why this research is conducted and what are the other areas that other researchers can work further in future.
(xiii) Should include more recent literature reviews between 2016 to 2021.
(xiv) Methodology of the work is well written.
.
(xv) Data collections was successfully conducted.
(xvi) Quantitative (statistical/qualitative analysis) is good. No further action needed.
(xvii) This section presents the results of the experiment and attempt to interpret their meaning.
(xviii) Do not present the raw data that you collected, but rather you need to summarize the data with text, tables and/or figures. This could be further improved.
(xix) Use the text of this manuscript to state the results of your study, then refer the reader to a table or figure where they can see the data for themselves.
(xx) Do not include the same data in both a table and a figure.
(xxi) Conclusions show readers the value of your completely developed argument or thoroughly answered question.
(xxii) Consider the conclusion from the reader's perspective.
(xxiii) At the end of this manuscript, a reader wants to know how to benefit from the work you accomplished in your manuscript.
(xxiv) Discussion made should be based on the objectives that had been reached. Can be further improved.
(xxv) References as mentioned should focus more references between 2016 to 2021. Otherwise looks good
Reviewer 2 Report
The paper reports the results of preparing the covalent organic frameworks – polyacrylonitrile electrospun nanofibers for dispersive solid-phase extraction of quinolones antibiotics from honey and pork for their subsequent analysis by the high-performance liquid chromatography technique. The authors have thoroughly investigated the influence of various parameters on the effectivity of dispersive solid-phase extraction using the prepared materials as sorbents. The prepared sorbents were characterized by utilizing the SEM, FTIR, XRD, and TGA techniques and the nitrogen adsorption at 77 K. Generally, the manuscript is well-written using good English. I recommend the manuscript be accepted for publishing in Nanomaterials after the authors clearly address the following issues:
1. Line 21: Please, decipher the acronym HPLC for the broad audience of the journal Nanomaterials.
2. Lines 22, 23, 25-27, and subsequently: Please, use dashes and minuses instead of hyphens if applicable here and throughout the manuscript.
3. Line 116: Please, provide more details for the SEM, FTIR, XRD, and HPLC measurements. Provide also the wavelengths used to record the chromatograms.
4. Line 121-122: Please, describe the methods used to calculate the surface area and pore size values using the measured isotherms.
5. Line 123: Is it correct that the desorption temperature during the low-temperature nitrogen adsorption examination was 100 degrees Celcius?! Check it, please, rigorously.
6. Lines 129-130: It is an isocratic regime, not gradient, as indicated in line 129 because the mobile phase composition was constant. The word “total” should be replaced with “volumetric”.
7. Lines 136, 177: Please, add the missed spaces.
8. Line 173: I recommend replacing the current version of this fragment with the following variant: “The dSPE procedure utilizing the PAN/TpPa-1 sample as an adsorbent was as follows”. I guess that it would be more appropriate.
9. Line 183: Please, check the English language grammar. I guess that “are” is more suitable than “were” here.
10. Line 235: I recommend replacing: “…cannot adsorb nitrogen…”
11. Lines 246, 338: Please, remove the excess spaces after the slashes.
12. Lines 260, 262: The word “elute” is a verb. So, I deem that the “eluent” or “eluant” would be more applicable.
13. Lines 312-313: I recommend replacing: “…for the subsequent HPLC determination…”
14. Line 346: Probably, “…for extraction…” would be more suitable.
15. Lines 347-348: I recommend the following variant of the sentence: “The prepared PAN/TpPa-1 material exhibited a good extraction effect towards quinolone antibiotics due to pi-pi-interaction and hydrogen bonds formation.” However, this statement should be proved, or the word “probably” should be added.
Round 2
Reviewer 1 Report
Accepted